



# Accounting for the effect of forest and fragmentation in probabilistic rockfall hazard

Camilla Lanfranconi[1], Paolo Frattini[1], Gianluca Sala[1], Davide Bertolo[2], Juanjuan Sun[3], Giovanni Battista Crosta[1]

[1] Università degli studi di Milano – Bicocca, DISAT, Dept. of Earth and Environmental Sciences, Milano 20126, Italy
[2] Regione autonoma Valle d'Aosta, Struttura attività geologiche, Quart 11020, Italy
[3] Key Laboratory of Shale Gas and Geoengineering, Institute of Geology and Geophysics, Chinese Academy of Sciences, Beijing 100029, China

*Correspondence to*: Camilla Lanfranconi (c.lanfranconi2@campus.unimib.it)

**Abstract**

The presence of trees along the slope and block fragmentation at impact strongly affect rockfall dynamics, and hazard as a consequence. However, these phenomena are rarely simulated explicitly in rockfall studies. We performed rockfall simulations by using the 3D rockfall simulator HY-STONE, modelling both the presence of trees and fragmentation through specific algorithms implemented in the code. By comparing these simulations with a more classical approach that attempt to account 15 implicitly for such phenomena in the model parameters, and by using a new probabilistic rockfall hazard analysis (PRHA) method, we were able to quantify the impact of these phenomena on the design of countermeasures and on hazard.

We demonstrate that hazard changes significantly when accounting explicitly for these phenomena, and that a classical implicit approach usually overestimates both the hazard level and the 95th percentile of kinetic energy, leading to an oversizing of mitigation measures.

**1. Introduction**

Rockfalls are widespread in mountain ranges, coastal cliffs, volcanos, riverbanks, and slope cuts, and threat people, structures and infrastructures, and lifelines (Crosta et al., 2015). Although rockfalls generally have a limited size, they are extremely rapid processes that exhibit high kinetic energies, long runout and damaging capability (Corominas et al., 2017). Rockfall hazard and risk assessment (Corominas et al., 2005; Agliardi et al, 2009; Lari et al, 2014; Wang et al., 2014; De Biagi et al, 25 2017; Farvacque et al, 2019, 2021; Hantz et al., 2021) and the design of defensive works (Volkwein et al., 2009) require numerical modelling of rockfalls to assess the dynamics of the blocks (i.e., velocity, kinetic energy and bouncing height) and the lateral and longitudinal spreading (Agliardi & Crosta 2003). In Italy, for example, the design of rockfall barriers is based on the use of the 95th percentiles of the blocks' height in flight and their kinetic energy, obtained from numerical models (UNI 11211; Volkwein et al., 2011). Since rockfall dynamics depends on block geometry, slope topography, surficial geology, 30 vegetation, and some peculiar rockfall behaviours (e.g. dynamic fragmentation), the reliability of analyses and the efficiency





of rockfall protections depend on the accuracy of modeling predictions and on the correct account for all relevant methodological issues (Crosta et al., 2015). Both the characteristics of the slope (e.g. topography, material properties and presence of forests) and the type of rockfall (e.g. whether it is fragmental) must be taken into account during modelling, because they contribute to the overall extent of rockfall potential and hazard zonation in mountain areas (Frattini et al, 2012). Both these characteristics can modify the trajectories, the extent and the dynamics of the rockfall events, the frequency, and the probability of impact.

Forests provide important protection against rockfall in steep mountain terrain, defending structures and infrastructures (Berger et al. 2002; Dorren et al. 2004a; Perret et al. 2004). Thanks to this nature-based solution, maintenance and installation costs of technical protection measures, such as embankments or nets, are financially bearable or can even be avoided at many places due to the reduction of rockfall rebound heights and impact energies by previous impacts on trees (Grêt-Regamey et al., 2008; Häyhä et al., 2015; Getzner et al., 2017; Moos & Dorren, 2021). Although this protective effect is evident in hazard assessment processes because it supports decisions on risk prevention measures, it is often accounted only in implicit terms, by adopting a set of modified restitution coefficients (Pfeiffer and Bowen 1989; Azzoni et al, 1995). More rarely, the presence of trees is simulated explicitely by using numerical modelling approaches (Dorren et al. 2006; Stoffel et al. 2006; Berger and Dorren 2007; Bigot et al. 2009; Jancke et al. 2009; Rammer et al. 2010; Leine et al. 2014; Radtke et al. 2014; Kajdiž et al. 2015; Dupire et al. 2016; Moos et al. 2017; Toe et al. 2018).

When stiff and strong rock blocks hit a hard impact substratum or other blocks of comparable size like a talus deposit, they may fragment and explode (Crosta et al., 2015). The rockfall fragmentation process is defined as the separation of the initial rock mass into smaller pieces generally upon the first impact on the ground (Evans & Hungr, 1993), and the resultant fragments propagate downslope following independent trajectories and new dynamics (especially in terms of kinetic energy and height) compared to the source block. This definition covers both the disaggregation of the block fragments delimited by pre-existing fractures in the initial mass and the generation of new fragments due to the breakage of intact rock (Corominas et al. 2012; Ruiz-Carulla, 2018). Block fragmentation is generally at the origin of extreme behaviors, major damages and accidents, and can interact strongly with protection structures (Nocilla et al., 2009; Wang et al., 2010; Corominas et al., 2019). Even if fragmentation during rockfall is recognized as fundamental in risk analysis (Corominas et al. 2012), a complete understanding of the process during rockfall has not been achieved so far, remaining a phenomenon largely neglected during numerical modelling. Only a few numerical codes allow modelling propagation that explicitly takes into account fragmentation (Crosta et al., 2003; Frattini et al, 2012; Matas et al., 2017; Ruiz-Carulla, 2018). When missing an explicit algorithm, the modeling of rockfalls with fragmentation can be done with two alternative approaches: either the model is calibrated to replicate the spreading of the event, including the most distal fragments, or the model is calibrated to replicate only the main deposit, neglecting the most distal blocks. The first approach leads to hazard overestimation, the second to hazard underestimation.

The aim of this paper is to quantify rockfall hazard when accounting for the presence of trees and fragmentation with an explicit simulation approach (i.e. using specific algorithms), and to evaluate the differences with a classical approach that does not simulate explicitly such phenomena. The simulator Hy-Stone (Crosta et al. 2003; Crosta et al. 2004), which allows to model



both the presence of forest and fragmentation, and a new revised Probabilistic Rockfall Hazard Analysis (PRHA) are adopted
to quantify the impact of these phenomena on the design of countermeasures and on hazard.

## 2. Methods

### 2.1 Rockfall simulations

#### 2.1.1 Hy-Stone

The simulation of rockfall propagation was performed by means of Hy-Stone, a 3D rockfall simulator that reproduces the
block motion from the dynamics equations (Crosta et al. 2004; Frattini et al. 2012; Dattola et al., 2021) using a triangulated
vector topography derived from Digital Terrain Models (DTMs). The stochastic nature of rockfall processes and parameters
are accommodated by slope morphology and roughness, and by the random sampling of most parameters from different
probability density distributions (e.g. uniform, normal, exponential). The block trajectories are computed by splitting them in
a succession of elementary motions: free fly, rolling, sliding and impacts/bouncing. When the impact process is concerned,
Hy-Stone has many different models comprising the constant and not-constant restitution coefficients (Pfeiffer and Bowen
1989) and the evolution of the elasto-visco-plastic model initially proposed by di Prisco and Vecchiotti (2006) and further
extended to prismatic blocks (Dattola et al., 2021). Specific model components explicitly account for the interactions between
blocks and countermeasures or structures, between blocks and trees, and fragmentation (Frattini et al. 2012).

#### 2.1.2 Tree-impact algorithm

The block-forest interaction is modeled through a stochastic tree-impact algorithm. Trees height, trunk diameter, absorbable
energy, and density (as number of trees in 10 meters square) are used by the algorithm to calculate at each cell a probability of
impact, that depends on the tree density, tree size and block size, and, in case of impact, a loss of energy and a lateral deviation
of the trajectories (Frattini et al., 2012) are considered. The energy lost by impact on tree stems is greatest for central impacts,
and decreases according to a Gaussian distribution away from the stem axis, while the angular deflection of the block on impact
is assumed to vary according to the type of impact (central, lateral, scour) (Dorren et al 2004b).

#### 2.1.3 Fragmentation algorithm

Hy-Stone can simulate the splitting up of a block in fragments moving independently from each other. The fragmentation
occurs at impact when the energy of a block exceeds a threshold defined by the relationship elaborated by Yashima et al,
(1987). The threshold fragmentation energy depends on the geomechanical properties of the block and its volume (the larger
the block, the lower the fragmentation energy). Once the fragmentation criterion is satisfied, a distribution of fragments is
generated with a size determined according to a power-law distribution. The number of fragments is computed according to a
mass conservation criterion, and the energy of each fragment is calculated assuming that the kinetic energy available after



impact is equally distributed among all fragments. Fragment projection velocity and direction are then computed according to energy conservation criteria. Results show that block fragmentation has an effect on the runout extent and on the spatial distribution of velocities and heights of the flying rocks. The largest fragments, however, display a behavior that is more similar to that of the parent blocks.

## 2.2  Rockfall hazard assessment

To assess rockfall hazard, we propose a new revised PRHA (Probabilistic Rockfall Hazard Analysis), based on Lari et al. (2014), to build rockfall hazard curves starting from a set of block-volume scenario simulations. For each block-volume scenario, the probability of exceeding a certain value of intensity (i, i.e. the reach of a specific value of kinetic energy), for each position along the slope (z) is:

$$P(I > i) = \int_{I_c}^{\infty} p(I)dI \qquad (1)$$


where $p(I)$ is the probability density function of kinetic energy at the position $z$. Multiplying the exceedance probability by the annual frequency of occurrence ($f$), we obtain the annual rate at which $i$ is exceeded, $F(I>i)$ as:

$$F(I > i) = f \cdot P(I > i) \qquad (2)$$

The annual frequency of occurrence ($f$) of each scenario combines the onset frequency ($f_o$) and the transit frequency ($f_t$) at a certain position:

$$f = f_0 \cdot f_t \qquad (4)$$

The onset frequency ($f_o$) of blocks with a certain volume, V, can be expressed in terms of magnitude frequency relationships
(Hungr et al. 1999; Dussauge et al. 2003; Rosser et al., 2007).

$$F = N(V) = aV^{-b} \qquad (5)$$

where $N(V)$ is the cumulative number of individual blocks with volume larger than $V$; $a$ depends on both the area extent and the overall susceptibility of the cliff, whereas the power law exponent, $b$, mainly depends on lithology and geological structure
(Hungr et al., 1999). To properly account for the frequency of individual blocks that propagate on the slope, it is necessary to combine the volume frequency relationship of rockfall events with the volume frequency relationship of blocks (Hantz et al., 2018; Hantz et al., 2020). The first relationship can be developed from surveyed historical events (e.g. Dussauge-Peisser et al. 2002; Chau et al., 2003; Guzzetti et al., 2003; Guthrie and Evans, 2004; Malamud et al., 2004) and provides annual frequencies of released rockfall volumes. However, these volumes should not be used for hazard analysis because single rockfall events



disaggregate or fragment (Ruiz-Carulla et al., 2017) soon after the detachment and during propagation into a distribution of smaller individual blocks. On the other hand, the volume frequency relationship of blocks can be derived from the rock mass fracture network or directly from already stopped blocks, both in the talus (Ruiz-Carulla et al., 2017), along roads (Hungr et al. 1999), and caught by rockfall nets within a certain range of time (Matasci et al., 2015; Moos et al., 2018). However, these distributions usually lack the temporal frame that allows to correctly estimate the annual frequency. The combination of the

two distributions can be achieved by calculating the total volume of the event (integrating the first distribution) and by calculating the $a$ parameter of the second distribution by assuming the total volume to be equal to the first one (Hantz et al., 2018).

The transit relative frequency ($f_t$) can be calculated for the rockfall simulation and corresponds to the ratio between the number of potential paths passing through a position ($t$) and the total number of simulated paths from the rockfall trajectories, ($t_{tot}$):


$$f_t = (\frac{t}{t_{tot}}) \qquad\qquad\qquad (6)$$

by assuming a homogeneous, stationary Poisson process for the occurrence of the events (Crovelli, 2000), the probability of exceeding each intensity i in the next T years from this annual rate, $P_{poiss}$, is:

$$P_{poiss}(I > i, T) = 1 - e^{-F_{tot}T} \qquad\qquad (8)$$


This represents the hazard curve at each position along the slope.

With respect to Lari et al. (2014), the revised PRHA method adopts a more flexible non-parametric approach for the kinetic energy probability distribution. Moreover, the new PRHA implements the approach proposed by Hantz et al (2016, 2019) for the calculation of the onset frequency ($f_o$), using the frequency-size distribution of the blocks observed along the talus to

downscale the magnitude-frequency distribution of larger study areas.

## 2.3 Demonstration case studies

The application of potential rockfall scenarios was performed at the two representative sites that were recently affected by rockfall events in the Aosta Valley Region (Western Italian Alps) showing a significant role of forest and fragmentation at

Saint Oyen and Roisan (Figure 1), respectively. During both the events, the rockfalls impacted roads and buildings, thus requiring a practical implementation of hazard assessment (for zonation) and the design of protection barriers (for mitigation). Saint Oyen and Roisan are located in the Western Alps, within the Austroalpine-Pennidic collisional prism, consisting of overburden layers formed by continental crust and fragments of oceanic lithosphere, strongly reworked by the Alpine tectono-metamorphic processes (Dal Piaz et al., 2016).



In the Saint Oyen case study (45°48'59.0"N 7°12'21.0"E), about 17,500 m³ of Ruitor micascists detached in March 2020, and reached a service road and the playing field in the lower part of the slope, passing through a mature fir forest. The presence of the forest significantly influenced the blocks distribution along the slope, increasing the lateral dispersion of trajectories and reducing their mobility. The case study is well documented by UAV flights conducted by the Regional Authority soon after the events, allowing for a detailed mapping of arrested blocks on the slope. We adopted this case study to investigate the role

of forest, which has been fundamental for the rockfall dynamic, as observed in the field. Although minor fragmentation may have occurred during the event, we neglected it during the simulation to focus on tree-impact only.

Less than 10 km far, at Roisan (45°47'49.3"N 7°18'49.0"E), about 1,050 m³ of Arolla gneiss toppled in October 2019 and impacted after 20 m of free fall (Polino et al., 2015) against a bench. While the main body of the rockfall stopped in a relatively flat area close to the source area, two blocks reached the foot of the slope causing the interruption of a municipal road. The

event is documented by a post-event UAV flight, and by a detailed field survey of the blocks. For this case study, the presence of forest was minor due to the size and age of the trees and it has been neglected in order to reveal better the role of fragmentation.

## 3.   Analysis and results

For both case studies, we firstly back-calibrated the model parameters on the rockfall events in order to simulate several volume

scenarios from local-scale rockfall source areas (with and without the use of specific algorithms for tree-impact and fragmentation) to quantify the differences in terms of dynamics, spreading and rockfall hazard. We simulated both all the scenarios by using the available 1 x 1 m Lidar DTM of Aosta Valley Region. The characteristics of each simulation, the number of simulated blocks, and the parameters adopted when using the two algorithms are reported in supplementary Table S1, 2 and 3.

**3.1  Calibration by back-analysis**

The calibration of model parameters was obtained by fitting the longitudinal and lateral extent of rockfall events by using the Hy-Stone model with and without tree-impact and fragmentation. In particular, we simulated the following scenarios (Figure 1):

-   SO_HS (Saint-Oyen tree impact implicit): the values of parameters are modified to account for the forest, e.g.
increasing rolling friction and reducing the tangential restitution coefficient. This is the most classical approach adopted in the practice to "simulate" the effect of forest with an implicit approach.

-   SO_HS$_{tree}$ (Saint-Oyen tree impact explicit): the values of parameters are calibrated by adopting the Hy-Stone tree-impact algorithm that explicitly simulates the effect of forest; in this case, the parameters used in the simulation do not account for the forest.



185    - R_HS (Roisan Fragmentation Implicit): the values of parameters are modified to allow the model to replicate the spreading of the event, including the most distal blocks, implicitly accounting for the possibility of fragmentation.

   - R_HS$_{frag}$ (Roisan Fragmentation Explicit): the values of parameters are calibrated by adopting the Hy-Stone fragmentation algorithm that explicitly simulates the distal blocks as fragments.

For Roisan, we experimented a different calibration strategy that replicates the spreading of the main deposit only (R_HS$_{short}$),
neglecting most distal blocks (Figure 1E). Although this strategy is physically correct to simulate non-fragmenting blocks, it provides an overall spreading that strongly underestimate the possible reach distance of fragments and the hazard level, accordingly.

For Saint-Oyen, both the simulations (R_HS, R_HS$_{tree}$) provide a good match with the main deposit of the 2020 event (Figure 1 A and B), with a slightly larger spreading when using the tree-impact algorithm, consistently with the fact that the impact
with trees adds a component of lateral dispersion to the trajectories.

For Roisan, we can observe a good match between the longitudinal and lateral extent of the main deposit from the 2019 event and the simulations (R_HS, R_HS$_{frag}$) but we observe an overestimation of the blocks reaching the paved road (18 blocks modelled, while just 2 blocks during the event) when the fragmentation algorithm is not used (Figure 1A). The comparison with simulated stopping points shows that the model without fragmentation is able to reach the maximum distance, but not in
the right location, since trajectories are strongly controlled by topography. This does not happen with the fragmentation algorithm, which is able to replicate the right position of the distal blocks in the meadow (Figure 1B).

Table 1 Table 2and Table 2 report the values of normal and tangential restitution coefficients and of the friction coefficient for the different slope materials used in the rockfall numerical simulations, and supplementary Table S1 and Table S2 the parameters of tree-impact and fragmentation algorithms.


### 3.2 Effect of tree-impact and fragmentation algorithm on kinetic energy

To quantify the effect of explicitly simulating tree-impact and fragmentation in rockfall modelling, we performed simulations for five volume scenarios (Table 3), using the modeling parameters that were back-calibrated from the events as previously described. For the spatial analysis, we divided the slope into a 10 x 10 meters square lattice and we calculated statistics of
kinetic energy within each square.

#### 3.2.1 Effect of tree-impact algorithm

Figure 2Figure 2 shows the 95[th] percentile of the blocks kinetic energy in each 10 m square. This statistic has been chosen
since it is frequently used for designing defensive works (UNI 11211; Maciotta et al., 2015; Lambert et al., 2021). In the case of small volume blocks, the simulation without tree-impact algorithm (Figure 2a) shows a central sector characterized by the highest kinetic energies (from 2,500 kJ up to over 10,000 kJ for the 95[th] percentile), and a distal zone characterized by lower





values. Trajectories are able to reach the base of the slope, the unpaved road, buildings, and playing field, and overpass the location of the outermost blocks of the 2020 event. When using the tree-impact algorithm (Figure 2B) the number of trajectories

passing through the central sector of the slope decreases dramatically. The trajectories that reach the base of the slope are concentrated in the area affected by the 2020 event where the forest is damaged. These trajectories reach only the unpaved road, with associated 95th percentile kinetic energy values of less than 2,500 J. For large blocks (Figure 2 C and D), the kinetic energy is high enough to nullify the effect of forest, and the two scenarios without and with tree-impact algorithm become similar.

From these results, it is evident that the use of the tree impact algorithm is relevant in the case of small volume blocks, for which the simulated trees are able to interrupt most of the computed trajectories, and in any case to decrease the kinetic energies. On the contrary, tree-impact simulation is almost irrelevant for large-volume blocks.

Analyzing the distribution of kinetic energies along the road at the foot of the slope without (HS) and with (HS$_{tree}$) the tree impact algorithm, we systematically observe lower values of energy for the HS models (Figure 3). In these models, the effect

of the forest is simulated by modifying the restitution and friction coefficients, calibrated on the range of kinetic energies of the event. However, this modification is independent of the kinetic energies of the simulated blocks, and it is not possible to observe the scale effect.

When the kinetic energies are lower than the calibrated kinetic energies and the kinetic absorption energies of the trees (S1 and S2), the classical HS approach overestimates the runout (see the large number of blocks intersecting the road after crossing

the forest in Figure 2 A). Instead, the HS$_{tree}$ algorithm intercepts, slows, and stops the least energetic blocks, allowing only the most energetic to reach the lower part of the slope. As a result, few transits are obtained, but with much higher kinetic energies due to the filtering effect of the forest (Figure 3).

When the kinetic energies grow beyond the calibration range (scenarios S4 and S5), the classical HS approach continues to apply the forest effect (through the modified parameters) even though the kinetic energies are well above the tree absorption

energies, underestimating the runout (the number of blocks intercepting the road remains about the same as in the low energy scenarios) and the kinetic energies (Figure 3). In contrast, the HS$_{tree}$ algorithm in these scenarios S4 through S5, shows higher kinetic energies and a high number of transits (compared to the lower-volume scenarios) because the effect of the trees becomes negligible, as it should be (Figure 3).

In the S3 scenario, greater congruence between the two approaches HS and HS$_{tree}$ is observed (Figure 3) because the simulated

volumes are more coincident with the calibration range (between 0.001 m$^3$ and 34 m$^3$).

### 3.2.2 Effect of fragmentation algorithm

Figure 4 shows the 95th percentile of the blocks kinetic energy in each 10 m square with and without the adoption of the fragmentation algorithm. The behavior of the models with small or large volumes is extremely different. In the case of small

volume blocks, the adoption of fragmentation algorithm is almost negligible, because blocks are too small to undergo fragmentation. In Figure 4 A the highest 95th percentile values of kinetic energy are concentrated in the area located just below





the modelled source and at the highest escarpment, and only four trajectories characterized by values up to 3 kJ reach and cross the paved road. In Figure 4 B we observe that the highest 95th percentile values are concentrated in the area close to the wall, but only one trajectory passes the road, characterized by 95th percentile of kinetic energy much lower (up to 1.5 kJ).

For larger blocks (S5 scenario), the difference is much more significant because more than half of the blocks are fragmented (612 out of 2646, 23%). In Figure 4C the runout achieved by blocks does not exceed that of Figure 4a, but with much larger values associated with the 95th percentile of kinetic energy reached all over the slope. The area located just below the modelled source and in the highest escarpment is characterized by kinetic energy values greater than 50,000 kJ at the intersection with the unpaved road. Values remain high also at the intersection with the paved road. In Figure 4D there is an increase in the

number of blocks crossing the roads, a consequent spread of trajectories with longer runouts (more than those actually achieved during the event) and a decrease in kinetic energy due to block fragmentation. On the unpaved road, the values associated with the 95th percentile drop to 50,000 kJ, and where the event boulder stopped it decreases to 8,000 kJ. At the intersection with the paved road, percentile values are more frequently lower than 15,000 kJ except in an isolated section where they reach 50,000 kJ and over.

Analyzing the distribution of Kinetic Energies along the paved road at the foot of the slope without (HS) and with (HS$_{frag}$) the fragmentation algorithm, we systematically observe higher values of energy for the HS models (Figure 5). Although during the event very few blocks crossed the paved road and only two of them reached the meadow at the foot of the slope, the calibration of the model without fragmentation was accomplished by adjusting the parameters in order to reach the maximum runout. This causes a strong overestimation of the number of blocks crossing the paved road, a general overestimation of the

landslide runout, and therefore also an overestimation of the kinetic energies at the element at risk. As already said for the Saint-Oyen case study, the runout in the HS models is almost independent from the kinetic energy of the blocks. Therefore, the number of transits is roughly constant in all six scenarios.

In the HS$_{frag}$ approach, the kinetic energy at the element at risk is systematically lower because the model is calibrated in order to allow only trajected fragments (characterized by much lower volumes with respect to original blocks) to reach and cross the

paved road as occurred during the event. The number of fragments reaching the road increases significantly through different volumes scenarios (from S1 to S5). This depends on the relationship between block size and fracture energy (Yashima et al., 1987); according to this relationship, the fracture energy scales with the radius of the block by an exponent that depends on the Weibull's coefficient of uniformity, and is always lower than 3, which is the scaling of the kinetic energy with radius. Hence, the larger the block, the higher is the probability of fracturing for a certain velocity.

The two approaches HS and HS$_{frag}$ provide similar results in the S4 scenario (characterized by simulated volumes that are more coincident with the calibration range, between 0.5 m$^3$ and 23 m$^3$) both in terms of number of blocks intersecting the road and in terms of kinetic energies: compared to all other scenarios, less than an order of magnitude separates the two 95th percentile values of kinetic energy.






## 3.3 Rockfall Hazard

The assessment of rockfall hazard requires the onset frequencies $f_0$ for each magnitude scenario, the transit frequency, $f_t$ and the distribution of kinetic energy in each position along the slope.

For the onset frequency (eq. 5), we adopted the methodology of Hantz et al (2018) to obtain the frequency distribution of

blocks for different volume classes, by combining the magnitude frequency relationship of rockfall events with the size frequency relationship of blocks along the talus. The first was obtained by analyzing the available rockfall database of the Valle d'Aosta region, which includes 306 events with volume information (Figure 6). Among them, only 25 belongs to the same catchment of the case studies (Buthier catchment, Figure 6). This subsample appears to be insufficient to characterize the magnitude frequency curve, especially for smaller volumes, and we therefore adopted the entire inventory that we fitted

with a maximum likelihood approach, obtaining a good power-law fitting ($R^2 = 0.99$) for rockfalls larger than 10 m³, with a scaling exponent of 0.56. We believe that this value is reliable also for the Buthier catchment, since the fitting curve has the same slope of larger rockfalls (with a volume greater than 500 m³) within the subsample. The size frequency relationship of blocks along the talus was obtained by image analysis only for Saint-Oyen due to a larger number of blocks and a better quality of the imagery. Figure 7 shows an excellent power-law fitting ($R^2 = 0.99$) for blocks larger than 0.2 m³, with a scaling

exponent $b$ equals to 1.22. Eventually, by combining the two distributions and accounting for the potential unstable area of both case studies, we obtained an $a$ value of 0.0072 and 0.0021 for Saint-Oyen and Roisan, respectively. The resulting onset frequencies for the different volume scenarios are reported in Table 3 for both case studies.

Both transit frequency ($f_t$) and the distribution of the kinetic energy come from the rockfall simulation trajectories sampled within 10x10 m cells. In order to characterize the kinetic energy distribution, we tested the hypothesis adopted by Lari et al,

(2014), who assumed the logarithm of the kinetic energy to be normally distributed, and obtained the kinetic energy probability density $p(I)$ by using the mean and standard deviation statistics. The Kolmogorov-Smirnov's test (Figure 8) shows that the normality is not rejected for more than 50% of the 10x10 m cells when using Hy-Stone without additional algorithms. However, this percentage is lower when using the tree-impact and fragmentation algorithms, suggesting that a non-parametric approach should be adopted when the level of complexity increases.

By combining the various scenarios and taking into account their associated probabilities, we constructed the hazard curves (by equation 8), which show the probability of exceeding a certain level of intensity in 50 years. Figure 9 Example of hazard curves characterized by a non-logarithmic trend, calculated in five cells of R_HS model.
  shows that hazard curves do not always have a logarithmic distribution, and that some curves do not reach the exceedance probability of 0.1 due to a very low transit frequency. Then, for each location along the slope and for each model run (SO_HS,

SO_HS$_{tree}$, R_HS, and R_HS$_{frag}$), we reduced each hazard curve to a single value in order to represent the hazard through a hazard map. As done by Lari et al (2014), we chose the kinetic energy with a 10% chance of being exceeded in 50 years, and we extracted this value from each hazard curve.

Compared to the SO_HS model, in SO_HS$_{tree}$ the hazard decreases because kinetic energy is significantly lowered, except in correspondence of the two sectors most affected by the event (see calibration in Figure 1) where it remains similar (Figure 10



A and B). The total area involved remains about the same, although with slightly lower runout. However, if only the areas with Ek>1 kJ are considered, hazard decreases significantly along the road at the foot of the slope.

For the Roisan case study, compared to the R_HS model, in R_HS$_{frag}$ the hazard decreases because the kinetic energy is significantly lowered, but note that the area involved increases (Figure 10 C and D). Analysing the distribution of the hazard values (Figure 11) at the foot of the slope obtained by the different approaches without and with the tree impact and

fragmentation algorithms, we observe an overestimation of the potential hazard in both case studies. In the Roisan case study, the overestimation is particularly high because the chance to fragment the blocks into smaller fragments greatly reduces the kinetic energy of those. Moreover, the distribution is less sparse because the only blocks with an energy value higher than the minimum energy value (1 kJ) that are able to reach the foot of the slope are few and localized in a 10-meter corridor.

## 4 Discussion

When hazard and risk need to be assessed, it is required to have a repeatable procedure and possibly a unique result. This study demonstrates that different modelling approach can influence the final result of hazard analysis and risk mitigation, but also points out the problems involved in advanced modelling, leading to necessary discussions on the topic.

### 4.1 Tree impact

The classical approach for modelling rockfall propagating along forested slopes is based on the modification of restitution and friction coefficients, calibrated on the extent of block propagation. This study shows that the adoption of this set of modified restitution coefficients provides a correct replication of the maximum lateral spreading and longitudinal runout, but inaccurate dynamic of blocks. In fact, the modification of the restitution coefficients is independent on the size of the blocks and can slow down even those blocks that are large enough to be actually unaffected by the presence of the forest. This leads to an

overestimation of rockfall runout and of the number of blocks reaching the elements at risk. When the protective role played by the forest is explicitly simulated (HS$_{tree}$), the hazard decreases due to the forest protection, but the high-percentiles of kinetic energy become higher. This occurs because the trees intercept the blocks with lower kinetic energy, with a *filtering effect* of the larger blocks, leading to the risk of considering, paradoxically, the presence of the forest as more dangerous. These considerations open an important discussion on the opportunity to design the defensive works only based on percentiles of the

kinetic energy.

### 4.2 Fragmentation

In case of rockfalls charcaterized by dynamic fragmentation, the classical approach for calibrating the model with this events is based on a conservative adjustment of the parameters in order to reach the maximum runout of single fragments. We

demonstrate that this approach leads to a strong overestimation of the number of transits (Figure 4), the overall landslide runout, the kinetic energy of blocks impacting the elements at risk (Figure 5), and the hazard (Figure 10). On the other side, the alternative approach to replicate only the main deposit, neglecting the most distal blocks would result in an underestimation



of all these quantities (supplementary Figure S5). Therefore, regardless of whether deciding to simulate only the blocks that have stopped in the main deposit (Figure 1E) or to extend the trajectories to the maximum fragment extent (Figure 1C), this
study demonstrates that the result is fundamentally incorrect, especially for the design of defensive works. On the other side, the explicit modelling of fragmentation is still challenging from both a theoretical point of view (Ruiz-Carulla et al., 2017; Shen et al., 2017; Guccione et al., 2022) and a practical point of view, due to the difficulty to calibrate the geotechnical parameters that control fragmentation. This adds further uncertainty in the analysis of rockfall dynamics and hazard.

From the results of hazard computation with and without fragmentation algorithm, which the hazard decreases. However, as
in case of forest, this results from the fact that more weight is given to the kinetic energy of the blocks and not the frequency. Is it correct to infer that the hazard decreases, even if the frequency increases, the kinetic energy of the blocks decreases, but the trajectories are dispersed much more? We believe that it is not completely correct because at spatial level, more zones are involved even if from smaller blocks. This discussion leaves room for future new studies.

### 4.3 Probabilistic rockfall hazard
The PRHA approach allows to quantify rockfall hazard in terms of hazard curves, thus describing the probability of exceeding a certain level of hazard. For each magnitude scenario, the approach overcomes the need of selecting a statistic of the kinetic energy at a certain positon along the slope (Agliardi et al, 2009; Farvacque et al, 2021), and allows to consider the full energy distribution within a certain grid cell. With respect to Lari et al. (2014), the revised PRHA method presents two improvements:
(i) it adopts a more flexible non-parametric approach for the kinetic energy probability distribution, instead of assuming a log-normal distribution, that we demonstrate in this paper to be frequently violated if tree impacts and fragmentation exist (Figure 8); (ii) it implements the approach proposed by Hantz et al (2016, 2019) for the calculation of the onset frequency ($f_o$). This approach allows to combine the onset frequency estimated from historical catalogues with the frequency size distribution of blocks along the slope. In fact, the large volumes recorded in the catalogues typically disaggregate into a population of blocks,
as soon as they impact on the slope. This disaggregation occurs due to the presence of pre-existing joints and fractures of a jointed rock mass (Ruiz-Carulla et al., 2017) and does not correspond to a dynamic fragmentation. The adoption of the Hantz et al (2016, 2019) approach place emphasis on the block size distribution along the slope, both to define the design volumes (Melzner et al, 2020), and to support the correct definition of the onset frequency.

### 5 Conclusions

The insight drawn from this study leads us to the conclusions that:
- If we do not explicitly simulate forest, we underestimate the protective role of trees and we consequently overestimate the hazard. On the other hand, the 95[th] percentile of the simulated kinetic energy of the blocks is higher when adopting the tree-impact algorithm because of the filtering effect performed by trees.



- If we do not explicitly simulate the fragmentation phenomenon, we overestimate the hazard in terms of energy values, but we underestimate the spreading of blocks during the events. The 95[th] percentile of kinetic energy along the element at risk is significantly lower when adopting the fragmentation algorithm.

- We obtained non-log-normal distributions of the kinetic energy values, so we adopted a non-parametric approach that we demonstrate being suitable for the hazard analysis. We highlight how PRHA fits different methodological models, and we quantify how much explicitly simulating both the interaction with forest and the fragmentation process lead to more accurate hazard mapping.

   As already mentioned in the discussion, we pointed the need to simulate a distribution of blocks that is representative of what already occurred as the so far most likely, because the dimensioning of the mitigation works is centered on the expected and simulated kinetic energies of the blocks. We also used the frequency-size distribution of the blocks along the talus to downscale the magnitude-frequency distribution of the study area, as proposed by Hantz et al (2018), to simulate different volume class scenarios.

- This study highlights the strong dependency of the 95[th] percentile of kinetic energy on the adopted modelling approach, showing the fluctuations of this value and thus the uncertainty related to the use of this parameter for hazard analysis.

## 6 Acknowledgments

The authors would like to acknowledge the support of the Geological Survey of the Aosta Valley Region, and also Serena Lari for useful discussions.

## 7   Data availability

The digital elevation model are publicly accessible at https://geoportale.regione.vda.it/download/dtm/ (Geoportale Regione Valle d'Aosta). The numerical simulations results are available from the first or corresponding author upon request.

## 8   Author contribution

CL, PF and GBC contributed to the conceptualisation of the project, formal analysis, investigation, methodology, visualisation, writing of the original draft, and reviewing and editing of the text. DB contributed by data collection with the Geological Survey of the Aosta Valley Region. GS and JS contributed to the analysis, reviewing, and editing of the text.

## 9   Competing interests

The authors declare that they have no conflict of interest.



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



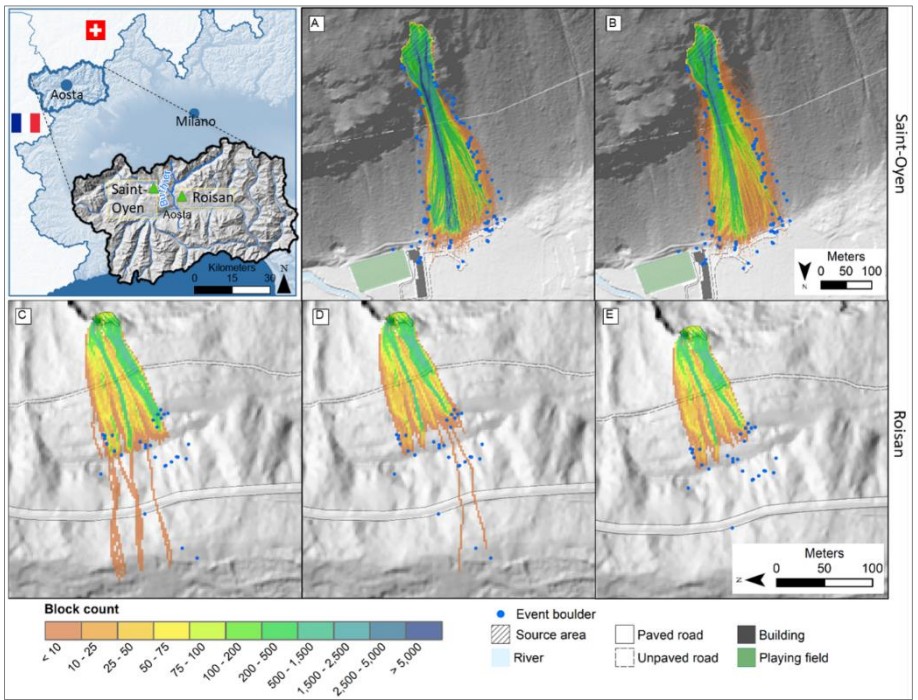

*Figure 1 A) Location of the two case studies in Aosta Valley region. The other panels show the back calibration of the rockfall events: A) and B) simulation of Saint-Oyen rockfall (A) with parameters modified to account for the forest (SO_HS) and (B) by adopting the Hy-Stone tree-impact algorithm (SO_HS_{tree}); C) and D) simulation of Roisan rockfall (C) with parameters*

*modified to implicitly account for the possibility of fragmentation (R_HS) and (D) by adopting the Hy-Stone fragmentation algorithm (R_HS_{frag}). Panel E) shows the calibration R_HS_{short}, obtained by neglecting the most distal blocks: this approach simulates only the blocks that stopped in the main deposit, without crossing the paved road.*





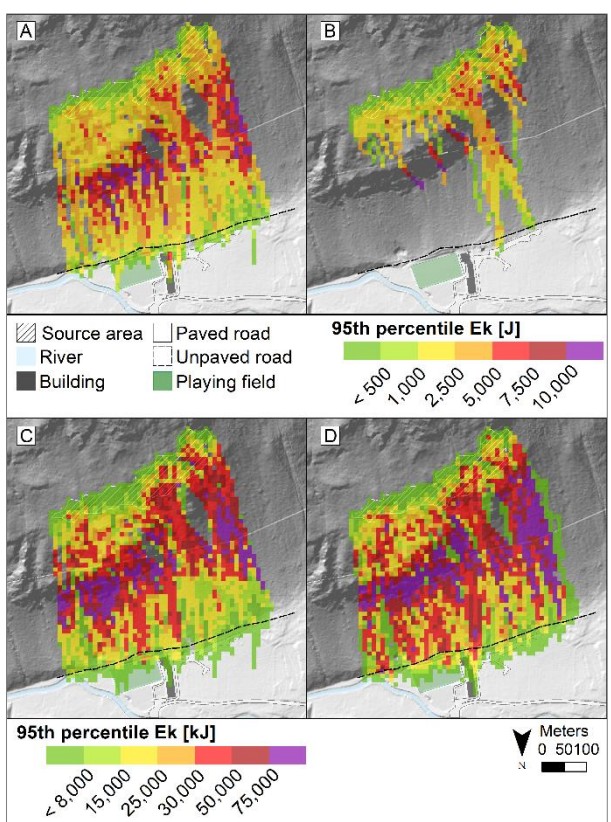

*Figure 2 Distribution of kinetic energies of blocks along the slope in Saint-Oyen case study. The value of each cell corresponds to the 95th percentile of the kinetic energy of the blocks passing through that cell. Box A) scenario S1 (small blocks) HS, B) scenario S1 (small blocks) HStree, C) scenario S5 (large blocks) HS, D) scenario S5 HStree (large blocks).*





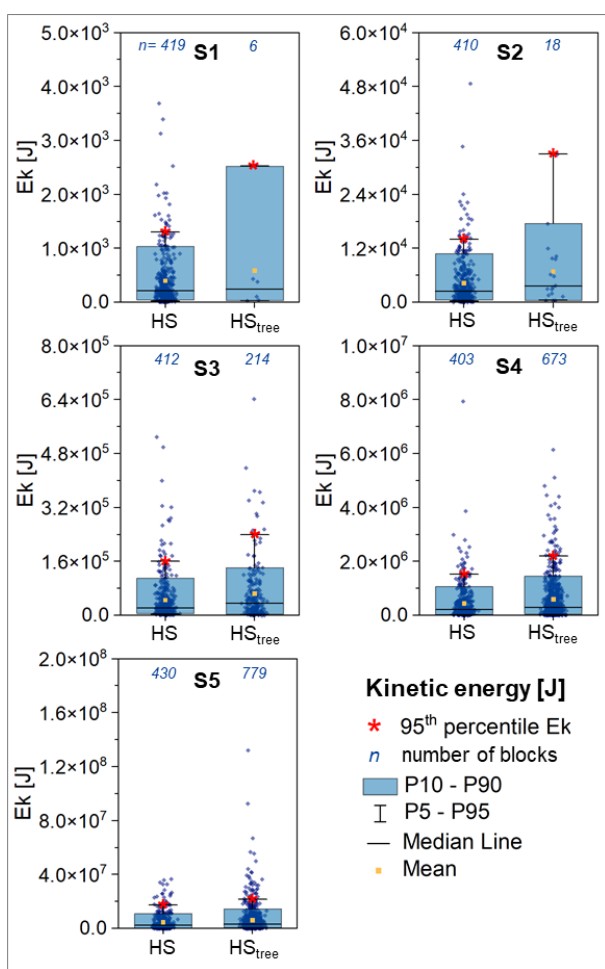

*Figure 3 Boxplots of kinetic energy values recorded for each scenario (S1 to S5) at the foot of the slope in Saint-Oyen case study. The associated 95th percentile value is highlighted by the red star. The total number of simulated blocks for each scenario is 995.*



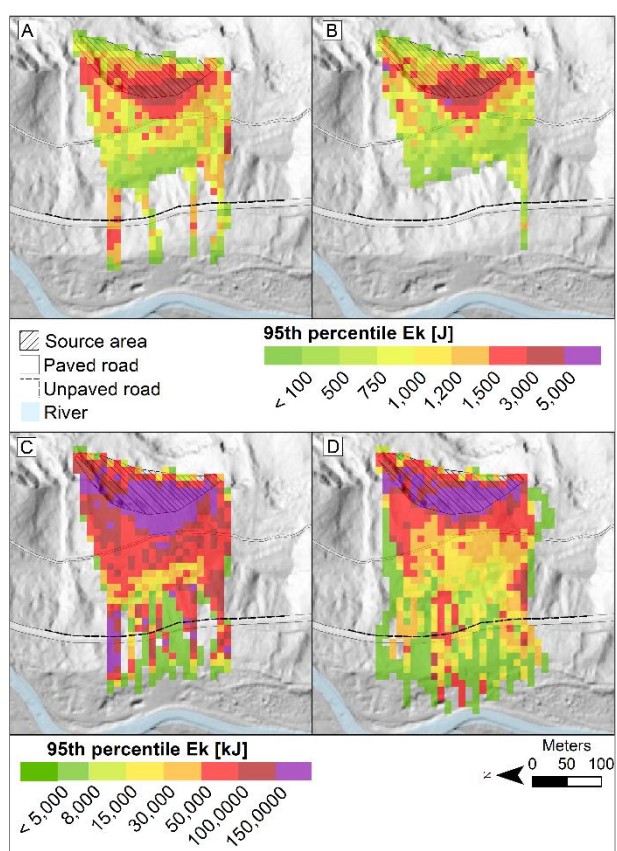

*Figure 4 Distribution of kinetic energy of blocks along the slope in Roisan case study. The value of each cell corresponds to the 95th percentile of the kinetic energy of the blocks passing through that cell. Box A) scenario S1 (small blocks) HS, B) scenario S1 (small blocks) HS_{frag}, C) scenario S5 (large blocks) HS, D) scenario S5 (large blocks) HS_{frag}.*





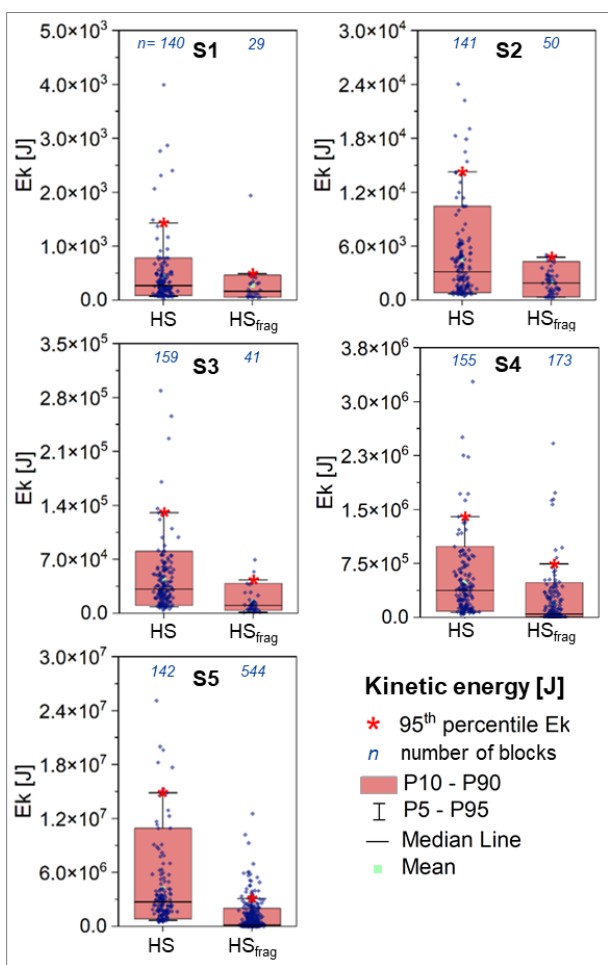

*Figure 5 Boxplots of kinetic energy values recorded along the road at the foot of the slope in Roisan case study. The associated 95th percentile value is highlighted by the red star. The total number of simulated blocks is 2646.*



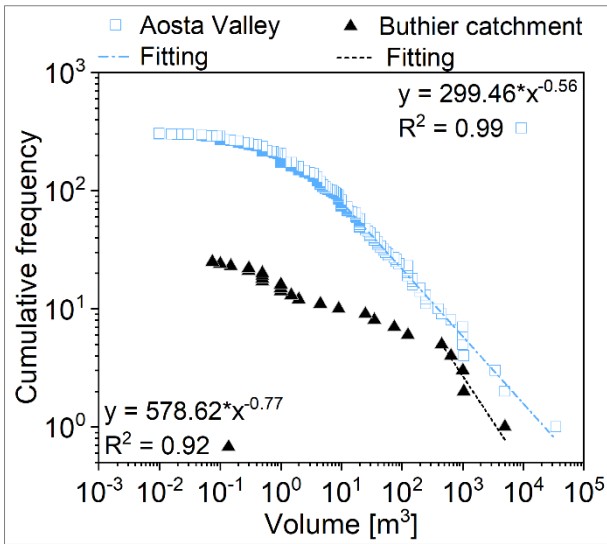

Figure 6 The two magnitude frequency relationships of 306 rockfall events collected in the Aosta Valley region (blue empty squares) and the 25 events from the Buthier catchment (black triangles).

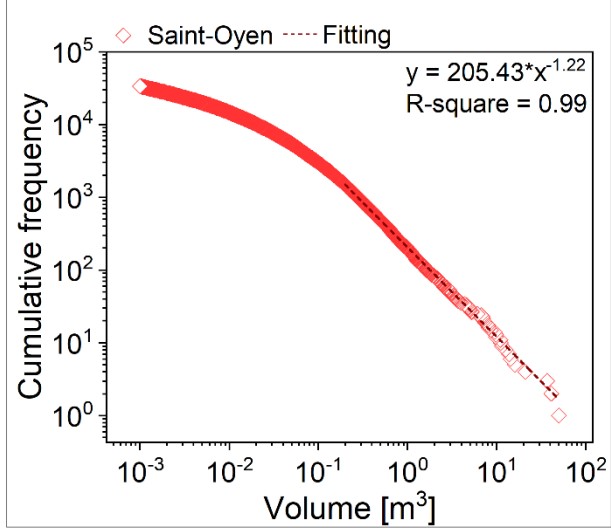

Figure 7 The size frequency relationship of blocks along the talus obtained by image analysis for the Saint-Oyen event.




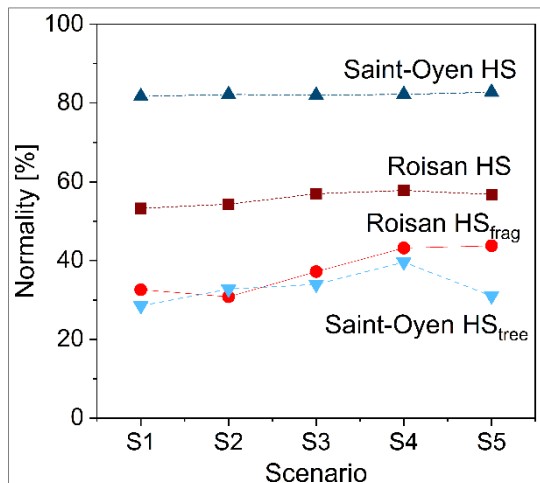

*Figure 8 Test of the normality of log-kinetic energy distribution within 10x10 m cells for all the volume scenarios. The y-axis shows the percentage of cells where the normality is not rejected by the Kolmogorov-Smirnov test.*

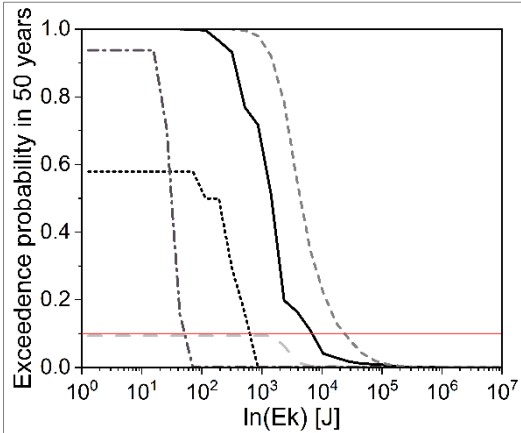

*Figure 9 Example of hazard curves characterized by a non-logarithmic trend, calculated in five cells of R_HS model.*



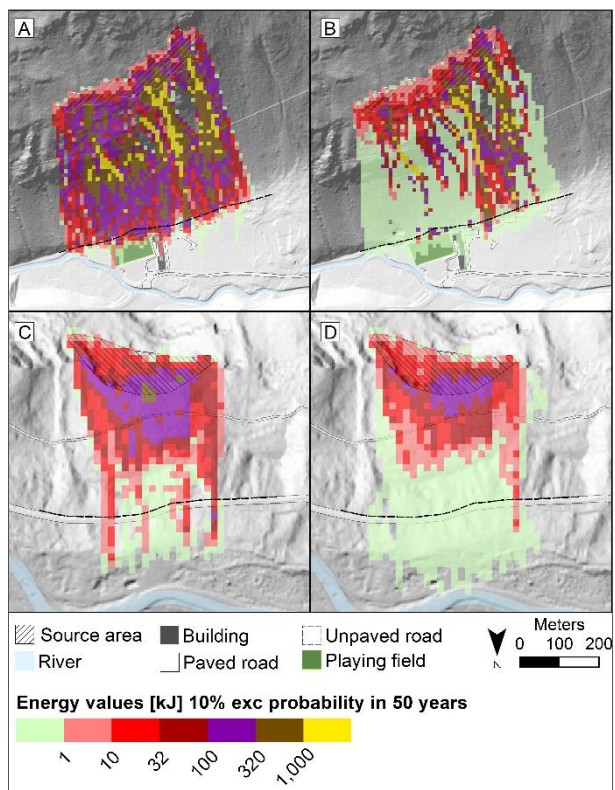

*Figure 10 Hazard map for: A) Saint-Oyen SO_HS model, B) Saint-Oyen SO_HS$_{tree}$ model, C) Roisan R_HS model, and D) Roisan R_HS$_{frag}$ model. The hazard is quantified as the kinetic energy associated to a 10% probability in 50 years.*




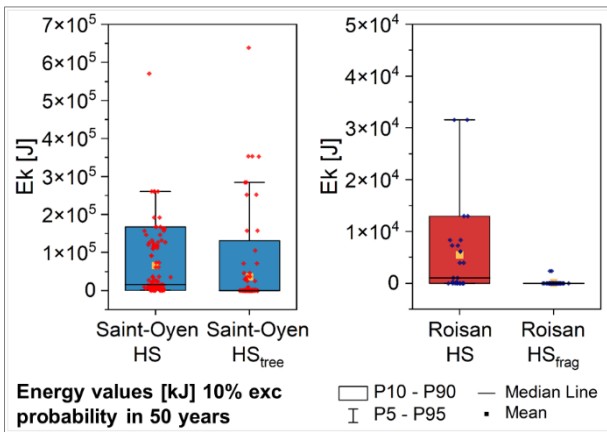

*Figure 11 Boxplot of the kinetic energy associated to a 10% probability in 50 years for Saint-Oyen (blue boxplots on the left)
and Roisan (red boxplots on the right) case studies, recorded along the road at the foot of the slope (dashed line in Figure 10).*

*Table 1 Values of normal ($e_n$) and tangential restitution ($e_t$) coefficients and of the friction coefficient ($\mu_s$) for the different
slope materials used in the rockfall numerical simulations for the Saint-Oyen case study.*

| Material | $e_n$ [-] | $e_t$ [-] | $\mu_s$ [-] |
|---|---|---|---|
| Outcropping rock | 85 | 85 | 0.3 |
| Coarse bare debris | 65 | 70 | 0.55 |
| Fine bare debris | 55 | 65 | 0.45 |
| Slope debris + damaged forest[1] | 64 | 71 | 0.5 |
| Slope debris + undamaged forest[1] | 75 | 80 | 0.4 |
| Alluvial deposit | 55 | 74 | 0.4 |
| Paved road | 70 | 77 | 0.3 |
| Unpaved road | 60 | 70 | 0.3 |
| Buildings | 20 | 10 | 1 |

*[1] Only for the explicit approach (HS_tree)*

*Table 2 Values of normal ($e_n$) and tangential restitution ($e_t$) coefficients and of the friction coefficient ($\mu_s$) for the different
slope materials used in the rockfall numerical simulations for the Roisan case study.*

| Material | $e_n$ [-] | $e_t$ [-] | $\mu_s$ [-] |
|---|---|---|---|
|  | HS | HS | HS |
| Outcropping rock | 75 | 85 | 0.2 |
| Sub-cropping rock | 60 | 70 | 0.3 |
| Slope debris in HS model | 60 | 65 | 0.4 |



| | | | |
|---|---|---|---|
| Slope debris in $HS_{frag}$ model | 50 | 60 | 0.5 |
| Paved road | 75 | 85 | 0.2 |
| Unpaved road | 55 | 65 | 0.3 |
| Alluvial deposit | 40 | 50 | 0.35 |


*Table 3 Volume scenarios for hazard analysis.*

| *Scenario* | **Range of volume [m³]** | **Roisan – onset frequency $f_o$** | **Saint-Oyen – onset frequency $f_o$** |
|---|---|---|---|
| S1 | 0.001 – 0.01 | 9 | 67 |
| S2 | 0.01 – 0.1 | 0.6 | 4.0 |
| S3 | 0.1 - 1 | 0.03 | 0.24 |
| S4 | 01-10 | 0.002 | 0.015 |
| S5 | 10 -100 | 0.0001 | 0.0009 |