# Peer review of "Accounting for the effect of forest and fragmentation in probabilistic rockfall hazard"

_EGUsphere, 2022_

## Author Response (AR1)

**Reply to Anonymous Referee #1**

*First, and regarding the fragmentation algorithm described in section 2.1.3, I suggest a more in detail explanation of each section of the algorithm. Even though some details on how fragmentation works on Hy-Stone can be found in Frattini et al. (2012), some parts of the process are not clear. Being the quantification of fragmentation effect one of the goals of this article I suggest a more technical description of the algorithm, including an explanation of how the Yashima et al. (1987) criterion is adapted for the fragmentation threshold and the criteria used to determine the post fragmentation directions (since this may control the spreading of the model).*

**Thank you for your comment. In the revised version, we added a more detailed section regarding the Hy-Stone fragmentation sub-model. Equations number 1 to 5 clarify the theory behind the model and the application of the Yashima et al. criterion.**

*Second, in the analysis of Roisan event, where fragmentation played a significant role as detailed in the manuscript, it would be very interesting to show the volume distribution of the deposited fragments in the field and compare it with the volumetric distribution obtained by the model during the calibration. If the model results correspond to the field measurements, the model calibration would be more complete regarding fragmentation. In this sense, it would also be convenient to explain what initial volumes are simulated to represent the event and how that initial distribution was determined. These suggestions could be included in section 3.1 if appropriate.*

**Thank you for your suggestions. For the revised paper, we prepared a figure comparing the volumetric distributions of the in-situ block size distribution (IBSD), the mapped block size distribution in the deposit (RBSD), and the simulated block size distribution with and without the fragmentation sub-model. The figure is now in the supplementary materials. Additionally, we included a brief explanation in section 3.1 about the initial volumes simulated and how they were identified.**

*Some minor comments:*

*Pag 5 line 133. For the calculation of the transit relative frequency when considering fragmentation, the total number of simulated paths include all generated fragments during propagation? When no fragmentation occurs this calculation looks trivial, but increasing the number of blocks during the propagation may lead to misuse of the classic approach.*

**Indeed, the comment is very interesting. In the calculation of the transit frequency, we use the number of simulated block before fragmentation. This may see counterintuitive because fragmentation can generate many blocks passing through a cell, and the value of frequency can potentially be larger than 1 (when the number of blocks passing through a cell is larger than the number of simulated blocks). This is correct, because this value is multiplied by the onset frequency, which is calculated without considering the dynamic fragmentation, and therefore it is consistent with the number of simulated blocks.**

*Pag 7 line 214: Typo: «Figure 2Figure 2»*

**Fixed**

*Pag 12 line 359: Is the sentence «From the results of hazard computation with and without fragmentation algorithm, which the hazard decreases.» correct? Can't get the meaning.*

**The sentence was incorrect. It has been fixed.**

*Pag 13 line 391: Possible bullet missing.*

**Fixed**

**Reply to Anonymous Referee #2**

*The manuscript represents a very interesting contribution to the understanding of the effect of forest and fragmentation in probabilistic 3D rockfall modelling. The rockfall model HY-STONE and the applied rockfall hazard methodology are valid and outlined clearly. However, the manuscript still lacks a bit of a clear structure, with in particular a sharp distinction between methods, results, and discussion. Furthermore, please check the required reference formatting and reference completeness text/list.*

**In the revised version, we attempted to make the chapter division more sharp. However, the overall paper is already divided into methods, analysis and results, and discussion. We checked the references carefully.**

*Some comments:*

*Line 80: please add if block volume & form are considered in Hy-STONE and how the forest/trees are included/considered in the model*

**Thank you for the comment. The Hy-Stone tree-impact sub-model simulates the presence of trees with a stochastic approach derived by Dorren et al. 2004. The reference is included in the methodology description. The volume and shape of the blocks are taken into consideration by Hy-Stone, and we added this information in the methodological chapter. Later, in the analysis chapter, we specified that we used spherical blocks for both case studies.**

*Line 155: HY-STONE is capable of simulating such large volumes?*

**Yes, Hy-Stone simulates individual blocks and it is able to compute the volumes described in the manuscript.**

*Line 214: typo "Figure 2Figure2"*

**Fixed**

*line 286: typo "Rockfall Hazard"*

**Fixed**

*Figures: please add the resolution of the data/simulations*

**As stated in the analysis chapter, we simulated all scenarios using the 1 x 1 m Lidar DTM of Aosta Valley Region. The parameters used with the two sub-models and the characteristics of each scenario are reported in supplementary Table S1, S2, and S3.**

**Reply to Editor**

*Dear Authors,*
*Referee reports on your paper indicate that only minor revisions would be required to accept your paper for publication in our journal. Reading your very interesting paper I agree with the reviewers that your paper must not necessarily be send out for another round of review. However, I made some additional observations listed below that you also should consider during revision.*

**Thanks, we considered all the observations very carefully. In addition, we further improved the manuscript to improve the quality of English and the reading comprehension.**

*I very much agree with the major comments of referee #1 concerning simulation of rock fragmentation and calibration of the algorithm.*

**We added a detailed description of the fragmentation sub-model in the manuscript to provide a description of the algorithm, also including new equations. We also decided to use "sub-model" instead of "algorithm".**

*P2L32: „all relevant methodological issues": what are these?*
**The sentence has been rephrased and the definition "methodological issues" removed.**

*P2L43: "more rarely": is this true given the large body of references cited here?*

**We agree that the general impression here is that many papers simulate tree impact explicitly. However, this is rare in the professional practice, and that was the reason for "more rarely". In any case, to avoid misinterpretations we substituted "more rarely" with "less frequently".**

*P4L100: Maybe a reference should be given that PRHA is initially based on the well-established concepts of PSHA?*

**We added the following sentences to line 100: "This methodology owes its idea on Cornell's (1968) probabilistic seismic hazard analysis (PSHA), which considers all possible earthquake scenarios to provide the exceedance probability of a certain level of ground motion at a site within a defined time frame."**

*P4L116: In Formula (5), shouldn't F be replaced by f0?*

**Fixed**

*P6L155: would be nice to have the coordinates also plotted in the maps in Fig.1*

**Added**

*P7L196-201: shouldn't rather Fig1C and Fig1D be referenced here?*

**Fixed**

*P7L201: Table 1 and Table 2?*

**Fixed**

*P9L253: A wall is not visible in Fig 4?*

**We meant the cliff. Fixed**

*P9L255: I think it would be nice to leave at least one sentence in the text explaining the magnitude scenarios considered*

**We have chosen these scenarios to encompass the size of blocks surveyed at Roisan and Saint Oyen. We added this sentence in chapter 3.2: "the volume scenarios range from 0.001 m3 to 100 m3 to encompass the block sizes surveyed on the field at the two sites."**

*P10L287-288 why and how?*

**We added "As explained in the methodology," to give a context where "why and how" are explained.**

*P10L310: how was the combination done?*

**A new equation (equation 12) was added in the methodology to explain the combination of scenarios, which is done by summing the values obtained for each scenarios for each cell of the grid. We added reference to Equation 12 in chapter 3.3.**

*P10L310-L317 Check this paragraph some broken sentence*

**Fixed**

*P11L331: is risk or its mitigation discussed anywhere in the paper?*

**The sentence was rephrased to be more clear, and "risk mitigation" was removed.**

*P12L359-364: This is all not clear, please rephrase and explain*

**Done**